# Genetically Encoded Fluorescent Sensors for SARS-CoV-2 Papain-like Protease PLpro

**DOI:** 10.3390/ijms23147826

**Published:** 2022-07-15

**Authors:** Elena L. Sokolinskaya, Lidia V. Putlyaeva, Vasilisa S. Polinovskaya, Konstantin A. Lukyanov

**Affiliations:** Center for Molecular and Cellular Biology, Skolkovo Institute of Science and Technology, Bolshoy Blvd 30, bld. 1, 121205 Moscow, Russia; elena.sokolinskaya@skoltech.ru (E.L.S.); l.putlyaeva@skoltech.ru (L.V.P.); vasilisa.polinovskaya@skoltech.ru (V.S.P.)

**Keywords:** COVID-19, SARS-CoV-2, coronavirus, protease PLpro, live cell imaging, genetically encoded probes, FRET, translocation

## Abstract

In the SARS-CoV-2 lifecycle, papain-like protease PLpro cuts off the non-structural proteins nsp1, nsp2, and nsp3 from a large polyprotein. This is the earliest viral enzymatic activity, which is crucial for all downstream steps. Here, we designed two genetically encoded fluorescent sensors for the real-time detection of PLpro activity in live cells. The first sensor was based on the Förster resonance energy transfer (FRET) between the red fluorescent protein mScarlet as a donor and the biliverdin-binding near-infrared fluorescent protein miRFP670 as an acceptor. A linker with the PLpro recognition site LKGG in between made this FRET pair sensitive to PLpro cleavage. Upon the co-expression of mScarlet-LKGG-miRFP670 and PLpro in HeLa cells, we observed a gradual increase in the donor fluorescence intensity of about 1.5-fold. In the second sensor, both PLpro and its target—green mNeonGreen and red mScarletI fluorescent proteins separated by an LKGG-containing linker—were attached to the endoplasmic reticulum (ER) membrane. Upon cleavage by PLpro, mScarletI diffused from the ER throughout the cell. About a two-fold increase in the nucleus/cytoplasm ratio was observed as a result of the PLpro action. We believe that the new PLpro sensors can potentially be used to detect the earliest stages of SARS-CoV-2 propagation in live cells as well as for the screening of PLpro inhibitors.

## 1. Introduction

The COVID-19 coronavirus pandemic from 2020–2022 has affected all areas of human activity; thus, the key role of fundamental research on animal and human viruses has become even more obvious. To date, more than 550 million cases of the disease have been recorded worldwide, more than 6 million of which have resulted in death. SARS-CoV-2 constantly mutates, resulting in new strains that often have much higher transmissibility; currently, Omicron is the dominating variant of SARS-CoV-2 [1,2]. Anti-COVID-19 vaccines designed for earlier coronaviral strains have turned out to be less effective for recent Omicron variants [3]. Thus, both the re-design of anti-COVID-19 vaccines and the development of new candidate viral inhibitors are in high demand. A detailed understanding of all stages of the virus–cell interaction can provide a solid basis for the development of effective diagnostic tools and drugs.

After binding to the ACE2 receptor on the cell surface, the virus introduces its genome into the cytoplasm. The viral genome consists of one positive-sense single-stranded RNA molecule about 30 kb long. The next step in the coronavirus life cycle is the translation of a so-called replicase gene from the 5′ part of the genomic RNA of the virion. The replicase gene encodes two large ORFs, rep1a and rep1b, which express two polyproteins, pp1a and pp1ab. The pp1a and pp1ab polyproteins contain nsp (non-structural proteins) 1–11 and 1–16, respectively, which are subsequently cleaved into separate nsps [4]. Coronaviruses encode two proteases that cleave replicase polyproteins: the papain-like protease PLpro, which is part of nsp3, and the serine protease Mpro, which is encoded by nsp5. PLpro cuts the boundaries of nsp1/2, nsp2/3, and nsp3/4; Mpro is responsible for the separation of the remaining 12 proteins. Proteases are the earliest viral enzymatic activities that trigger all subsequent events leading to virus replication and release. Importantly, viral proteases also cut many host proteins, a process crucial for the virus replication [5]. In particular, PLpro recognition sites are present in ubiquitin and ISG15 (interferon-induced gene 15), so PLpro possesses high de-ubiquitinating and de-ISGylating activities. The essential roles of viral proteases make them promising targets for the development of antivirals [6,7].

The PLpro domain of about 250 amino acids is a part of the large (~2000 residues) nsp3 protein, which includes many domains [8]. Importantly, nsp3 has four transmembrane segments and locates to the endoplasmic reticulum (ER) membrane (Figure 1). Most of the protein, including the PLpro domain, is exposed to the cytosol.

To quantify the activity of SARS-CoV-2 proteases by means of fluorescence, which is ideal for high-throughput screening, several assays have been developed [9]. The simplest in vitro test for purified proteases is based on recognition site-bearing short peptides labeled with two fluorophores in such a way that upon cleavage the Forster resonance energy transfer (FRET) between them disappears [10,11]. This assay enables the careful quantification of enzymatic activity. More complex cell-based models rely on the expression of the protease and corresponding fluorescent sensor in mammalian cell cultures. Although less quantitative than in vitro, such assays ensure the functioning of the protease in native intracellular conditions. For screening applications, it assesses not only protease inhibition, but also the membrane permeability and cell toxicity of the tested compounds.

Frogatt et al., developed a sensor for the screening of potential inhibitors for the main SARS-CoV-2 protease 3CLpro [12]. The fluorescent reporter is based on the FlipGFP protein, which becomes fluorescent only after beta-strand reorientation [13]. In the presence of an active protease, a linker with the cleavage site is cut; this leads to proper GFP conformation and fluorescence. In the presence of a known 3CLpro inhibitor, the fluorescence of the reporter was reduced as a result of protease inhibition. Recently, Ma et al., developed a similar FlipGFP-based sensor for the discovery of SARS-CoV-2 PLpro inhibitors [10]. A sensor was used for drug screening in a cell-based assay; two candidates out of the ten tested were chosen for a more advanced assay and showed cellular antiviral activity in micromolar concentrations. Both sensors can be successfully used in screening assays for testing new inhibitors of SARS-CoV-2 proteases.

Hahn et al., developed a FRET-based 3CLpro based on CFP and YFP fluorescent proteins linked by a 3CLpro cleavage site [14]. In the presence of an active protease, the FRET signal was disrupted by protein separation after proteolysis whereas the addition of the protease inhibitor led to the restoration of the FRET signal. The generated FRET sensor was tested on a known 3CLpro inhibitor in combination with the FlipGFP assay.

Here, we designed new variants of genetically encoded fluorescent sensors for the real-time detection of PLpro activity in live cells. We aimed to detect PLpro activity in individual live cells without the addition of external dyes, ensure the real-time monitoring of PLpro activation, use red-shifted fluorescent proteins to reduce phototoxicity, and build a flexible genetic system for the easy replacement of all modules for a specific task.

## 2. Results

### 2.1. Red FRET Sensor for Soluble PLpro

The first strategy for PLpro detection includes the expression of a soluble protease and soluble fluorescent sensor in the cytoplasm of mammalian cells. An nsp3 part, including the PLpro domain and the ubiquitin-like globular domain (UBL-2), was expressed under the control of an inducible promoter. To detect the entry of the plasmid into the cells and the induction of expression, the blue fluorescent protein mTagBFP2 [15] separated from the protease by the T2A peptide was used (for the synthesis of two individual proteins from one open reading frame).

A red-shifted genetically encoded sensor based on the FRET between a GFP-like red fluorescent protein as a donor and a biliverdin-binding near-infrared fluorescent protein as an acceptor was designed. We previously used this design for a caspase-3 sensor consisting of a FRET pair, mKate2 and iRFP, separated by a DEVD cleavage site [16]. For the new PLpro sensor (“mScarlet-LKGG-miRFP670”), we selected brighter fluorescent proteins; namely, mScarlet [17] and miRFP670 [18]. The PLpro recognition site LKGG [8] was introduced into the linker between them. Similar to many previous FRET-based protease sensors, we expected the disappearance of the FRET after the linker cleavage and, hence, an increase in the fluorescence intensity of the mScarlet donor (Figure 2A).

To test the PLpro sensor in mammalian cells, the following experiments were carried out. The PLpro-T2A-mTagBFP2 and mScarlet-LKGG-miRFP670 constructs were co-transfected into HeLa cells. The next day, after the accumulation of the mScarlet-LKGG-iRFP670 sensor, the PLpro-T2A-TgBFP2 expression was induced with doxycycline. Fluorescence was then monitored for several hours in two channels: red (mScarlet) and far-red (miRFP670). The signal in the blue channel (mTagBFP2) was assessed at the end of the observation to confirm the PLpro expression without blue light-induced phototoxic effects. The FRET efficiency was assessed by the ratio of the mScarlet/miRFP670 signals. This allowed us to avoid any artifacts of sensor accumulation, cell movement, and shape changes. Time-lapse imaging showed that the mScarlet/miRFP670 signal ratio gradually increased 1.3–1.6-fold for several hours after the induction of the PLpro expression (Figure 2B). At the same time, in the control cells not expressing PLpro, no significant increase in the mScarlet/miRFP670 ratio was observed (Figure 2B). Thus, we concluded that mScarlet-LKGG-miRFP670 provided a specific FRET-based assessment of the PLpro activity.

### 2.2. Translocation Sensor for Membrane-Associated PLpro

Although the PLpro domain itself is soluble, both the PLpro catalytic domain and its target cleavage sites are parts of the membrane protein. Thus, an association with the membrane may be important for protease activity, e.g., by the proper positioning and/or increase in the apparent protein concentration in the 2D membrane environment. To get closer to native conditions, we designed a system for the expression of the membrane-bound fluorescent sensor and PLpro.

We constructed a fusion of the green mNeonGreen and the red mScarletI fluorescent proteins, which are bright and capable of efficient expression and fast maturation in mammalian cells [17,19], connected by the PLpro cleavage site-containing linker (Figure 3A). At the C-terminus, this sensor also carried a short membrane “tail anchor” (TA) from the tyrosine phosphatase PTP1B, which ensures localization to the cytosolic face of the ER membrane [20,21]. The same TA localization signal was used to direct PLpro to the ER.

mScarletI-LKGG-mNeonGreen-TA showed an expected intracellular localization to the ER as a fine network in the cytosol in both the green and red channels (Figure 3B). Upon the induction of the PLpro-TA expression, the red fluorescence underwent a gradual change in its intracellular localization: the ER network disappeared and the red signal became evenly distributed throughout the cell (Figure 3B). At the same time, the green signal kept the distribution unchanged. We concluded that the observed changes fully corresponded with the expected PLpro-catalyzed cleavage of the sensor into the soluble mScarletI part and the ER-bound mNeonGreen-TA part. An intranuclear red signal appeared to be the simplest readout of the sensor as it was very low without PLpro and became high after PLpro activation. Thus, we calculated the nucleus/cytosol ratio in the red channel and normalized it to the nucleus/cytosol ratio in the green channel for the same regions of interest (ROIs). This value increased about two-fold in the cells after PLpro activation, providing a robust readout (Figure 3C).

## 3. Discussion

Genetically encoded fluorescent protein-based sensors are efficient tools due to their ability to work in live cells, high specificity, and easy targeting of any intracellular compartment [22]. In the present study, we constructed two sensors for the viral protease PLpro activity based on different readouts; namely, FRET and intracellular translocation. The choice of PLpro was made for the following reasons. Firstly, PLpro is the earliest viral enzyme, which is, therefore, critical for all subsequent steps of an infection. Secondly, it was shown that in addition to performing its main function, viral polyprotein processing, PLpro was involved in the inhibition of the host immunity response. The protease dysregulates interferon antiviral signaling and reduces inflammation by removing ISG15 and ubiquitin from cellular proteins [8]. Therefore, PLpro inhibitors can potentially not only suppress viral replication, but also restore the regulation of impaired signaling cascades in infected cells [10]. Thirdly, much more literature on Mpro sensors and inhibitors is currently available, making PLpro understudied.

The soluble cytoplasmic FRET sensor mScarlet-LKGG-miRFP670 showed about a 1.5-fold increase in the donor fluorescence intensity, which was similar to that for the analogous sensors of caspase-3 activity [16]. The ratiometric FRET imaging used here is convenient for the real-time monitoring of individual cells. At the same time, we thought that for the end-point analysis, fluorescence lifetime imaging microscopy (FLIM) of the mScarlet donor was more appropriate as it ensured the absolute quantification of the FRET. In contrast to a previously published Mpro sensor based on a classical CFP–YFP (cyan–yellow) FRET pair [14], we used red and near-infrared fluorescent proteins. The red-shifted spectra of the sensor made it possible to reduce the phototoxicity of the observations, which is important for time-lapse imaging. Moreover, mScarlet-LKGG-miRFP670 is highly suitable for multiparameter imaging together with green and blue channels. For example, a large collection of target proteins fused with GFP variants as well as GFP-based sensors are available [16,22]. In addition, endogenous cofactors NAD(P)H and FAD, which show the metabolic status of the cell [23,24], can potentially be imaged together with mScarlet-LKGG-miRFP670. 

The idea of a translocation sensor for PLpro is new. The ER-associated sensor mScarletI-LKGG-mNeonGreen-TA has the following main advantages. Firstly, it places PLpro and its target site into a more natural, membrane-bound environment. Secondly, the ER-to-nucleus translocation of mScarletI represents a clearly visible and quantifiable readout. Due to double-normalization—nucleus/ER ratios in the red and green channels and then a ratio of these ratios—the sensor response does not depend on the expression level and maturation rates of the green and red fluorescent proteins. Thirdly, it consists of very bright and fast-maturing fluorescent proteins of the last generation, which makes imaging easier for researchers and safer for cells. Fourthly, as mNeonGreen and mScarletI represent a good FRET pair [25], the sensor mScarletI-LKGG-mNeonGreen-TA can potentially be used in the FRET mode. However, the translocation of the acceptor protein makes it difficult to quantify the FRET changes by commonly used ratiometric calculations. FLIM of mNeonGreen would overcome this problem by detecting an increase in the donor fluorescence lifetime when the acceptor is cut off. Finally, both mScarletI-LKGG-mNeonGreen-TA and mScarlet-LKGG-iRFP670 did not show any cleavage by cellular proteases in the absence of PLpro. This suggests a high specificity of their response. 

Importantly, both our sensors ensured (nearly) the real-time detection of PLpro activation. Indeed, FRET changes occur instantly; the translocation of a monomeric fluorescent protein (mScarletI in our case) between the cytoplasm and nucleus takes a few seconds [26]. In contrast, FlipGFP-based protease sensors require a longer time (tens of minutes [13]) for GFP chromophore maturation.

For our sensor genetic constructs, we used the Golden Gate (MoClo) cloning system [27]. It enabled the fast replacement of the sensor modules—the fluorescent proteins, protease gene, and cleavage site—that provided an opportunity to change the sensor specificity or fluorescent properties. Thus, it could be easily adapted for different specific tasks such as an analysis of other viral or cellular proteases.

The present work utilized a simple experimental model of the co-expression of the sensors with a plasmid-coding PLpro. An important issue to be addressed in further studies is the sensitivity of the sensors during the infection of cells with the SARS-CoV-2 virus. Indeed, sensor overexpression can potentially result in too high an excess of the substrate over low amounts of PLpro produced from the viral genome; this reduces the sensitivity of the assay. Thus, the careful optimization of the expression level of the sensor is required to attain the maximal sensitivity and speed of detection of the viral infection. To this end, stable cell lines expressing sensors at different levels can be compared in terms of fluorescence response upon a viral infection. 

We believe that the new genetically encoded PLpro sensors are a potentially useful tool to detect the earliest stages of SARS-CoV-2 propagation in host cells as well as for the search of PLpro inhibitors in cell-based screening platforms.

## 4. Materials and Methods

### 4.1. General Methods and Materials

DNA oligonucleotides for the PCR were commercially synthesized by Evrogen (Moscow, Russia). An Encyclo PlusPCR kit (Evrogen PK101) was used for the routine PCR amplifications. ScreenMix-HS with HS Taq (Evrogen PK143S) was used for the screening of *E. coli* colonies after the transformation. The length of the PCR products was verified on 1.2% TAE agarose gels. The PCR products of interest were then cut and purified with a Cleanup S-Cap kit (Evrogen BC041S). The DNA plasmids were cultured overnight and then extracted using a Plasmid Miniprep kit (Evrogen BC021L). The gene sequences were verified by Sanger sequencing.

### 4.2. DNA Cloning

The DNA constructs were cloned using the Golden Gate cloning system [27]. The genes of interest were cloned into Level 0 and Level 1 backbones obtained from a MoClo Toolkit (AddGene Kit #1000000044). BpiI (BbsI) and Eco31I (BsaI) restriction endonucleases (Thermo Scientific, Waltham, MA, USA) and T4 DNA ligase (Evrogen LK001) were used for the MoClo cloning procedure. 

The expression vector encoding SARS-CoV-2 nsp3 was a gift from Fritz Roth (AddGene plasmid # 141257 [28]). PLpro-coding DNA fragments were generated by PCR amplification on the template of the nsp3 coding sequence for soluble PLpro (UBL-2 domain + PLpro catalytic domain, amino acid positions 747–1065) and for T-anchored PLpro (UBL-2 domain + PLpro catalytic domain + nucleic acid-binding domain, amino acid positions 747–1202) (see Figure 1 in [8]). The PLpro genes were put under the control of a TRE3G promoter (Tet-On^®^ 3G Inducible Expression System, Takara Bio Inc., Kusatsu, Shiga, Japan). Transfection efficiency was visualized using mTagBFP2 protein fused to the “self-cleaving” T2A peptide GSGEGRGSLLTCGDVEENPGP [29]. 

DNA constructs for the PLpro FRET sensor consisted of a donor protein (mScarlet) and an acceptor protein (miRFP670) connected by an amino acid linker GGGSGLKGGGGSGS, where LKGG corresponded with the PLpro cleavage site. 

DNA constructs for the PLpro translocational sensor consisted of two fluorescent proteins (mNeonGreen and mScarletI) connected by an amino acid linker GGGSTLKGGAPTKVGGSGS, where TLKGGARTKV corresponded with the nsp2–nsp3 junction from the SARS-CoV-2 polyprotein. 

All coding sequences together with the promoter and terminator were assembled into Level 1 expression vectors using the Golden Gate cloning system. DNA constructs encoding the PLpro gene were put under the control of an inducible promoter (TRE3G promoter) and assembled into a pLVT-like vector for further lentiviral production if needed. The DNA constructs encoding the PLpro FRET sensor and the translocational sensor were put under a constitutive CMV promoter and possessed a SV40 poly(A) sequence.

### 4.3. Cell Culture and Transfection

HeLa cells were cultured at 37 °C (5% CO_2_) in DMEM (PanEco, Moscow, Russia) supplemented with 10% fetal bovine serum (BioSera, Nuaille, France), 100 U/mL penicillin, and 100 mg/mL streptomycin (PanEco). For the live cell imaging experiments, the DMEM was replaced by imaging media: MEM (PanEco) supplemented with 10% fetal bovine serum (BioSera) and 20 mM HEPES (Corning, New York, NY, USA).

### 4.4. Live Cell Imaging

For the live cell imaging experiments, HeLa cells were seeded on glass-bottomed 35 × 10 mm dishes (SPL Life Sciences, Gyeonggi-do, Korea) and were incubated at 37 °C (5% CO_2_) overnight. The next day, the cells were transfected with ViaFect Transfection Reagent (Promega, Madison, WI, USA) according to the manufacturer’s protocol. For the experiments with the soluble PLpro FRET sensor, the cells were transfected with DNA constructs encoding the FRET sensor, PLpro gene, and Tet-On 3G transactivator protein at a 1:1:1 ratio (3 µg of DNA in total) in OptiMEM (Gibco, Thermo Scientific, Waltham, MA, USA) (1:3 DNA:ViaFect ratio). For the experiments with the translocational PLpro sensor, the cells were transfected with DNA constructs encoding the FRET sensor, PLpro gene, and Tet-On 3G transactivator protein at a 1:1:1 ratio (3 µg of DNA in total) in OptiMEM (Gibco) (1:3 DNA:ViaFect ratio). The imaging experiments were performed the next day using a BZ-9000 inverted fluorescence microscope (Keyence, Osaka, Japan). Prior imaging of the PLpro expression was induced by 10 µg/mL doxycycline (Sigma-Aldrich D9891, St. Louis, MO, USA). The imaging was carried out at 37 °C with a 60 × PlanApo 1.40 NA oil objective (Nikon, Melville, NY, USA) for 7–10 h with a 15 min frame interval. A TexasRed OP66838 BZ filter (Keyence, Osaka, Japan) was used to induce the mScarlet and mScarletI fluorescence, a GFP-BP OP66836 BZ filter (Keyence) was used to induce the mNeonGreen fluorescence, a 49022-ET-Cy5.5 filter (Chroma Technology, Bellows Falls, VT, USA) was used to induce the miRFP670 fluorescence, and a 49021-ET-EBFP2/Coumarin/Attenuated DAPI filter (Chroma Technology) was used to induce the mTagBFP2 fluorescence.

### 4.5. Image Analysis

The images were processed using Fiji ImageJ [30]. The time-lapse images were arranged into a stack. The quantification of the fluorescent signals was performed by measuring the fluorescence intensity in the circular ROI in red and far-red or red and green channels. The FRET efficiency was assessed by the ratio of the mScarlet/iRFP670 signals. Temporal changes to the ratio in each cell were normalized to the first frame corresponding with the PLpro induction point by doxycycline.

For the calculation of the nucleus/cytoplasm ratio, the mean fluorescence was measured in the nucleus and cytoplasmic ROIs, respectively; the same ROIs in the green and red channels were used for each individual cell. For all calculations, the mean fluorescence intensity of the background out of the cells was subtracted. 

## 5. Conclusions

We designed two new genetically encoded fluorescent sensors for the activity of SARS-CoV-2 protease PLpro, which is essential for the processing of the viral polyprotein as well as for the cleavage of many host proteins and the repression of the cellular antiviral response. The FRET-based sensor mScarlet-LKGG-miRFP670 enabled the real-time monitoring of the PLpro activity in individual cells. The red-shifted fluorescence of the sensor ensured a low phototoxic observation and was well-suited for multiparameter imaging, leaving the blue–green channels free. The translocation-based sensor mScarletI-LKGG-mNeonGreen-TA used a natural-like association of PLpro and the sensor with the ER membrane. It showed a fast redistribution of the red signal to the nucleus upon cleavage with PLpro. A simple ratiometric quantification of the signals in both the red and green channels ensured that the readout was independent of protein synthesis and maturation. Further experiments are needed to show the applicability of the new sensors for the detection of cell infections with a functional SARS-CoV-2 virus. The sensors can potentially be used to develop a safe virus-free cell-based platform for the screening of PLpro inhibitors.

## Figures and Tables

**Figure 1 ijms-23-07826-f001:**
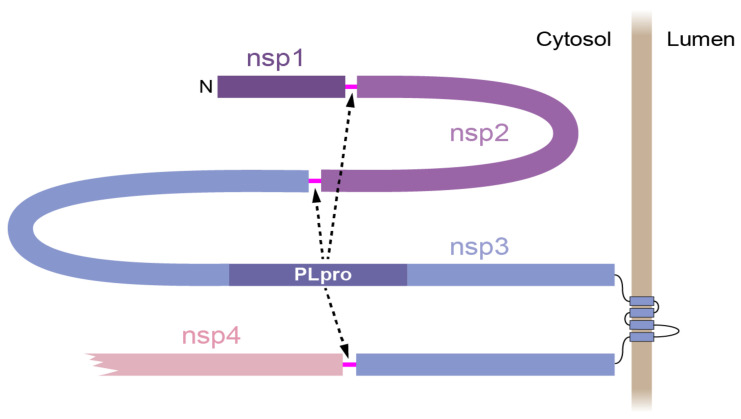
Schematic representation of an N-terminal part of the SARS-CoV-2 polyprotein ORF1a/1b. Nsp1, nsp2, and nsp3 with PLpro domain and four transmembrane segments (across endoplasmic reticulum membrane, shown as vertical brown bar) as well as a part of nsp4 are shown. PLpro cleavage sites (magenta) are between nsp1, nsp2, nsp3, and nsp4.

**Figure 2 ijms-23-07826-f002:**
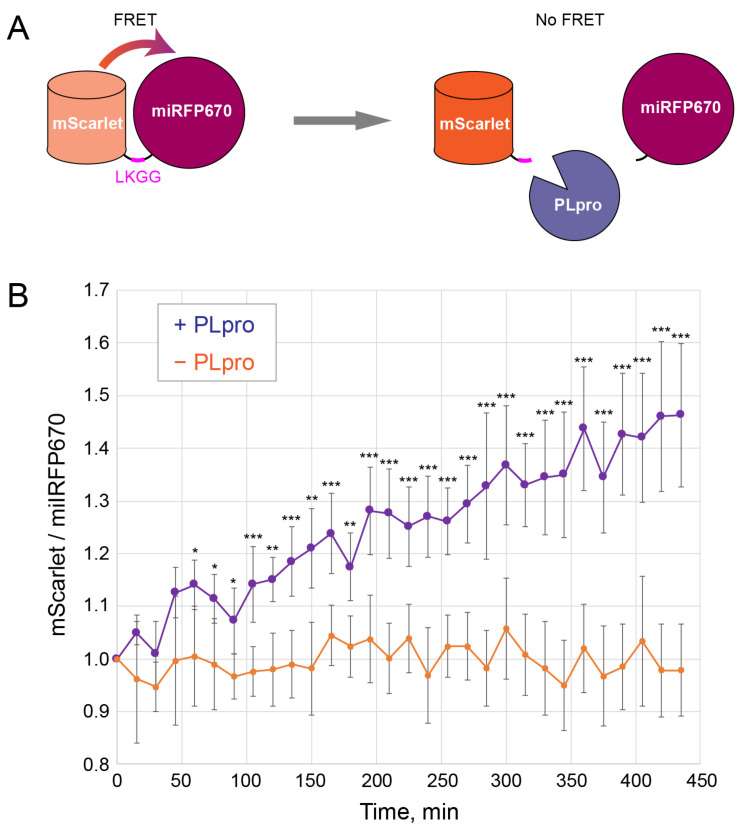
FRET sensor of PLpro activity. (**A**) Scheme of the sensor consisting of the FRET donor mScarlet and acceptor miRFP670 fluorescent proteins connected by the linker with LKGG recognition site for PLpro. PLpro cleaves the linker and thus eliminates FRET, leading to the increase in mScarlet fluorescence intensity. (**B**) Changes in the mScarlet/miRFP670 fluorescence ratio in cells during time-lapse microscopy upon induction of PLpro expression (purple) or without PLpro (orange). Mean and standard deviation for 15 cells in 3 independent experiments for each curve are shown. *, **, ***: *p*-values less than 0.05, 0.01, and 0.001, respectively (*t*-test).

**Figure 3 ijms-23-07826-f003:**
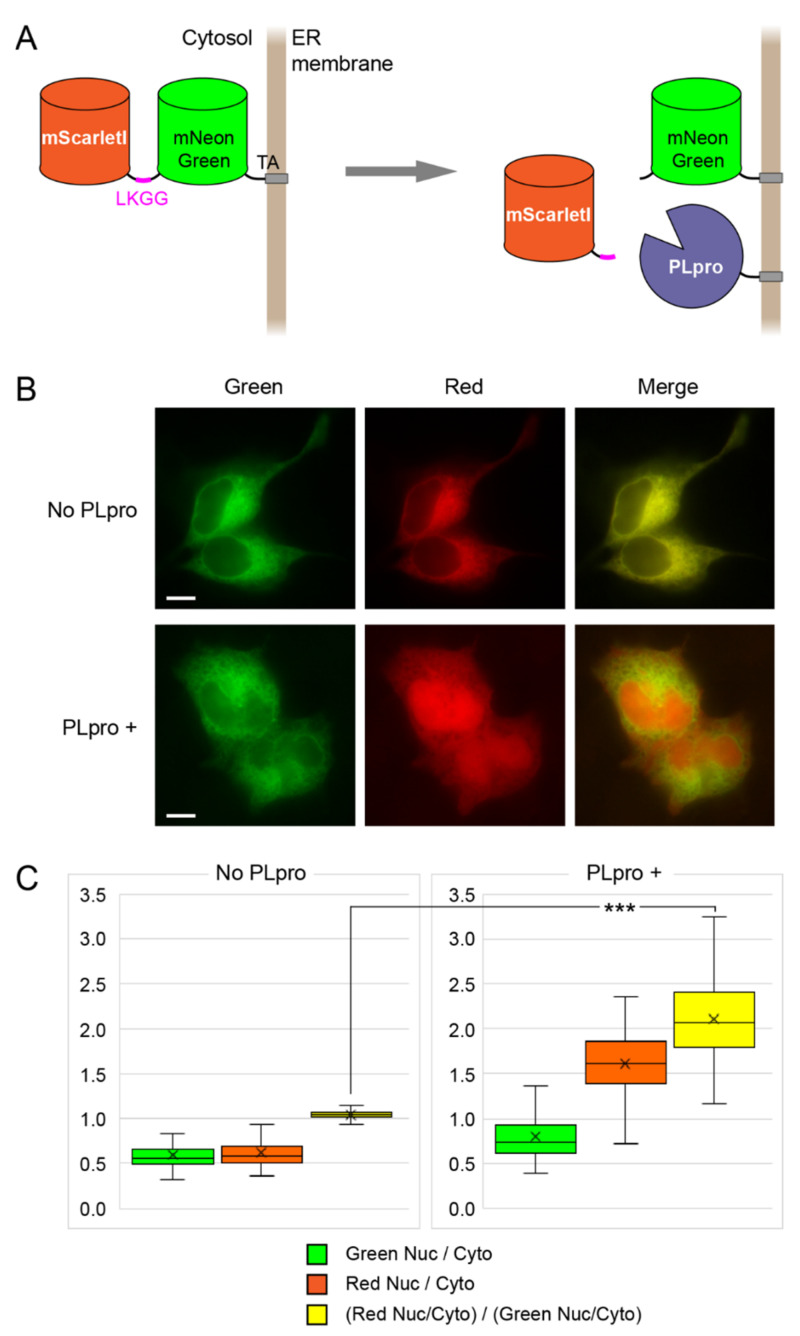
Translocation sensor of PLpro activity. (**A**) Scheme of the sensor consisting of the mScarletI and mNeonGreen fluorescent proteins connected by the linker with LKGG recognition site for PLpro and C-terminal TA localization signal. PLpro-TA (with the same ER-bound localization) cleaves the linker and thus liberates mScarletI from the ER surface. (**B**) Representative examples of cells expressing mScarletI-LKGG-mNeonGreen-TA in the absence (top) or presence (bottom) of PLpro-TA. Images in the green and red channels and their merging are shown. Scale bars: 10 µm. (**C**) Box and whisker plots for nucleus/cytosol ratios for cells expressing mScarletI-LKGG-mNeonGreen-TA sensor without PLpro (left, *n* = 190 cells) and in the presence of PLpro (right, *n* = 150 cells). Median, mean, 1st, and 3rd quartiles as well as minimum and maximum values are shown for each plot. The red and green boxes are the ratio of fluorescence in the nucleus to the cytoplasm in the corresponding channels and the yellow plot is the ratio of the obtained values to each other. ***: *p*-value less than 0.001 (*t*-test).

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
