# Peer review of "Genetically Encoded Fluorescent Sensors for SARS-CoV-2 Papain-like Protease PLpro"

_ijms, 2022, doi:10.3390/ijms23147826_

Round 1

Reviewer 1 Report

Sokolinskaya et al report on two new genetically encoded sensors for studying SARS-CoV-2 protease. One sensor is FRET-based, and the other is relocalization-based. Overall, the paper is well written, and data are clearly presented.

The following matters require some additional attention:

1)    In the introduction several existing sensors are described for monitoring the same SARS-CoV-2 protease activity. In the discussion, a bit more emphasis could be given in comparing the benefits of the new sensors to the already existing sensors.

2)    In lines 187-194 it is discussed that the second sensor based on mNeonGreen-mScarlet-I would also be a good FRET sensor, but that ratio imaging provides a problem by the relocalization of mScarlet-I after cleavage. I think this depends on how the ratio is calculated. If you divide the green signal by the red signal, a good readout would be expected by dividing increased  ER-localized green fluorescence  by less red fluorescence, and a non-FRET signal in the nucleus where you divide no green fluorescence by a lot of red fluorescence after cleavage. One would expect to start with an even ratio all over the cell followed by an increase in FRET-ratio near the ER and a decrease around the nucleus. That contrast change could be picked up independent from protein concentration.

3)    There is no discussion on sensitivity of the sensors. As seen in fig.2 the mScarlet/miRFP670 ratio gradually increases over a period of hours. Ideally the increase is only dependent on protease levels, yet I think the kinetics are also dependent on the amount of sensor produced: upon overproduction of the sensor it would need much more protease activity for a similar ratio change than in case of a modest production of the sensor. The question here is how the levels of co-expression found after co-transfection of the HeLa cells compare to the levels of production of sensor and of the protease in screening for the progress of a viral infection?

Reviewer 2 Report

The manuscript by Sokolinskaya et al. “Genetically encoded fluorescent sensors for SARS-CoV-2 pa-pain-like protease PLpro” described approaches for real-time detection of PLpro activity in live cell by two genetically encoded fluorescent sensors. The manuscript findings are interesting, and require revision to address a few concerns as follows:

Comments.

1. Lines 26-29, the paragraph may be elaborated with some more detailed pieces of information in brief (few sentences) on COVID-19 screening, transmission, mortality, treatment strategies, and recent trouble due to its variants: doi: 10.1016/j.ijsu.2022.106727; doi:10.1007/s12088-020-00893-4.

2. Lines 82-83, please add the main objectives and significance of the present study in a brief. 

3. Discussion is week. It should be minor improved in detail and justify the significance.

4. Section 4, the details of the instruments should be provided i.e. modal no., manufacturer, and county of origin.

5. Please add a conclusion section to highlight the significance of the finding and future directions.
